# CONTRASTE: Supervised Contrastive Pre-training With Aspect-based Prompts For Aspect Sentiment Triplet Extraction

**Rajdeep Mukherjee**     **Nithish Kannen**     **Saurabh Kumar Pandey**     **Pawan Goyal**
Indian Institute Of Technology Kharagpur
rajdeep1989@iitkgp.ac.in, nithishkannen@gmail.com,
saurabh2000.iitkgp@gmail.com, pawang@cse.iitkgp.ac.in

## Abstract

Existing works on Aspect Sentiment Triplet Extraction (ASTE) explicitly focus on developing more efficient fine-tuning techniques for the task. Instead, our motivation is to come up with a generic approach that can improve the downstream performances of multiple ABSA tasks simultaneously. Towards this, we present **CONTRASTE**, a novel pre-training strategy using **CONTR**astive learning to enhance the **ASTE** performance. While we primarily focus on ASTE, we also demonstrate the advantage of our proposed technique on other ABSA tasks such as ACOS, TASD, and AESC. Given a sentence and its associated (*aspect, opinion, sentiment*) triplets, first, we design **aspect-based prompts** with corresponding sentiments *masked*. We then (pre)train an encoder-decoder model by applying contrastive learning on the decoder-generated aspect-aware sentiment representations of the *masked* terms. For fine-tuning the model weights thus obtained, we then propose a novel multi-task approach where the base encoder-decoder model is combined with two complementary modules, a tagging-based *Opinion Term Detector*, and a regression-based *Triplet Count Estimator*. Exhaustive experiments on four benchmark datasets and a detailed ablation study establish the importance of each of our proposed components as we achieve new state-of-the-art ASTE results.

## 1 Introduction

ASTE is the most interpretable Aspect-based Sentiment Analysis (ABSA) task that, given a sentence, requires the extraction of *opinion triplets*, each consisting of an *aspect term*, the *sentiment* expressed towards it, and the corresponding *opinion term* explaining the rationale behind the sentiment, as shown in Table 2. Existing ASTE approaches, ranging from pipeline (Peng et al., 2020a), multi-task (Zhang et al., 2020), tagging-based (Xu et al., 2020; Wu et al., 2020), span-based (Xu et al., 2021; Feng et al., 2022), and generative (Mukherjee et al.,

2021; Chen et al., 2021; Yan et al., 2021; Zhang et al., 2021a) have all developed newer, better, and often more complex fine-tuning techniques for the task, gradually enhancing performance on the benchmark datasets (Xu et al., 2020). Our motivation instead is to essentially improve the aspect-aware sentiment understanding of a generative architecture that can simultaneously benefit multiple ABSA tasks. Towards this direction, we propose **CONTRASTE**, a **CONTRA**stive learning-based efficient pre-training strategy that enhances the downstream performance of **ASTE**. Although we primarily focus on ASTE, we also demonstrate the advantage of our proposed approach in achieving state-of-the-art performances for other ABSA tasks such as Aspect Category Opinion Sentiment (ACOS) quad prediction, Target Aspect Sentiment Detection (TASD), and Aspect Extraction and Sentiment Classification (AESC).

Supervised Contrastive Learning (SCL) (Khosla et al., 2020) has been previously explored in ABSA (Li et al., 2021b; Liang et al., 2021; Ke et al., 2021). SCL is performed on label (here sentiment) representations, whose distance in the embedding space, during training, is pulled together from the ones with the same sentiment orientation and pushed apart from those with different sentiment polarities. The existing works however differ from ours in their objectives and approaches. A detailed comparison is presented in Section 5. Most importantly, the existing techniques apply SCL on sentence-level sentiment embeddings, whereas in ASTE (or ABSA in general), the sentiments are defined at an *aspect level*. To address this limitation, we introduce **aspect-based prompts** with [MASK] tokens. A sentence, for e.g., *the food was delicious*, is sent as input to the **T5** (Raffel et al., 2019) encoder. The decoder is then provided with an aspect-based prompt, <aspect> *food* <sentiment> [MASK]. The decoded representation for the [MASK] token gives us the aspect-aware representation of the cor-

| Sentence | Pre-training Prompts
<aspect> *aspect* <sentiment> [MASK] |
|---|---|
| The food was good. | <aspect> food <sentiment> [MASK] |
| Both sound as well as display quality are great. | <aspect> sound <sentiment> [MASK]
<aspect> display quality <sentiment> [MASK] |
| While the sushi was tasty, the ambience sucked. | <aspect> sushi <sentiment> [MASK]
<aspect> ambience <sentiment> [MASK] |

Table 1: Few sentences along with aspect-based prompts derived from them to pre-train an enc-dec framework.

responding sentiment label. Aspect-level SCL is performed on these embeddings. Our proposed pre-training scheme is depicted in Fig 1(a). Few example sentences along with the aspect-based prompts derived from them to pre-train the framework are shown in Table 1. In Section 3.5, we compare the two pre-training approaches, that is performing SCL on sentence-level versus aspect-level sentiment embeddings, and demonstrate that the latter (our proposal) leads to better ASTE performance.

After pre-training is completed, we need to fine-tune the trained model weights for the downstream ABSA tasks (we focus on ASTE here). Similar to PARAPHRASE (Zhang et al., 2021a), the most competitive generative approach for ASTE and other ABSA tasks, we train our *base* encoder-decoder framework, **CONTRASTE-Base**, to generate opinion triplets as *templatized paraphrases*, as shown in Fig. 1(b). Our template however differs. We argue that the template used in PARA-PHRASE, although more intuitive, does not generalize well across tasks, and often results in semantically meaningless target sequences, as reported in Table 2. Taking inspiration from Huguet Cabot and Navigli (2021), we therefore design a more generic placeholder-based template that can be leveraged across ABSA tasks. Finally, we train *CONTRASTE-Base* with two auxiliary multitask objectives, *Opinion Term Detection* and *Triplet Count Estimation*, each of which benefits the ASTE performance. We call our final model **CONTRASTE-MTL**.
Our contributions can be summarized as follows:

- We propose a novel strategy to continually pre-train T5 for ABSA tasks (from its generic pre-trained checkpoint). More specifically, given a sentence, we derive (possibly multiple) *aspect-based prompts* with corresponding sentiments *masked* as shown in Table 1. The model is then pre-trained by applying supervised contrastive learning (SCL) on the decoder-generated *aspect-level* sentiment representations of the masked tokens as shown in Fig 1(a). We show that such an approach leads to better downstream ASTE performance than performing SCL on *sentence-*

*level* sentiment embeddings for pre-training, as in existing works. Also, different from prior works, we **do not use any additional data** to perform contrastive pre-training of our model.
- We propose more generic placeholder-based templates than the ones used in PARAPHRASE (refer to Table 2) to fine-tune T5 for ASTE. We further demonstrate that the template can be easily customized for various ABSA tasks.
- Our final model, *CONTRASTE-MTL* (Fig. 1(b)), pre-trained using SCL on *aspect-level* sentiment embeddings, and fine-tuned for ASTE using a multi-task objective, achieves state-of-the-art (SOTA) results on all four benchmark datasets.
- We also show that our proposed pre-training strategy helps in achieving SOTA results for other ABSA tasks such as ACOS, TASD, and ASEC.
- Finally, we compare our results for all four tasks against **ChatGPT** (OpenAI, 2023b), and observe substantial gains in performance.

Section 2 presents the details of our proposed pre-training and fine-tuning methodologies. Section 3.4 demonstrates the advantage of our placeholder-based templates, over the ones used in PARAPHRASE, in achieving better ASTE performance. Section 3.5 compares our pre-training strategy with the one used in prior works. Here, we visually demonstrate how pre-training on aspect-centric sentiment embeddings results in deriving more discernible clusters of representations with different sentiment polarities. Our main results are reported in Section 3.6. Section 4 discusses our model ablations and demonstrates how pre-training helps in improving the performances of other ABSA tasks, such as ACOS, TASD, and AESC.

## 2 Methodology

### 2.1 Supervised Contrastive Pre-training

We model ASTE as a structured prediction task, and leverage the **T5** (Raffel et al., 2019) encoder-decoder framework as the backbone of our proposed architecture. The empirical justification for this design choice primarily comes from prior

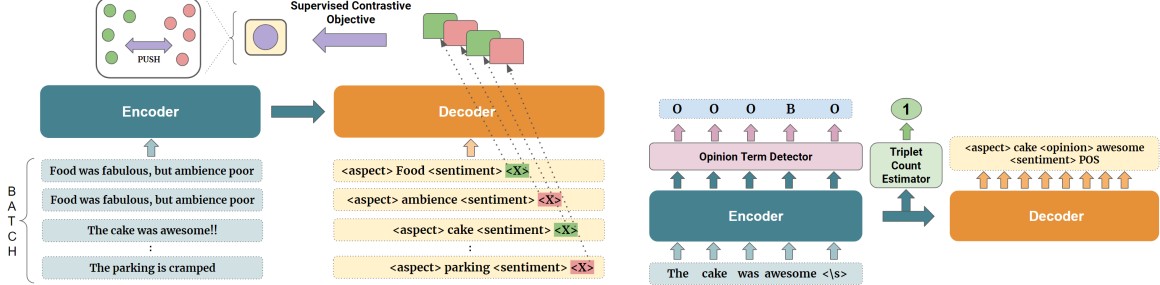

**(a) Supervised contrastive pre-training**   **(b) Proposed multi-task model for ASTE**

Figure 1: **CONTRASTE**: (a) Contrastive pre-training of the encoder-decoder framework using aspect-based prompts. (b) Fine-tuning the model for ASTE by optimizing a joint-objective to generate template-based triplets.

| SENTENCE | CONTRASTE template: <aspect> *aspect* <opinion> *opinion* <sentiment> *sentiment* [SSEP] ... | PARAPHRASE template: It is great / ok / bad because *ASPECT* is *OPINION* |
|---|---|---|
| The food was good. | <aspect> food <opinion> good <sentiment> POS | It is great because food is good |
| While the sushi was tasty, the ambience sucked. | <aspect> sushi <opinion> tasty <sentiment> POS [SSEP] <aspect> ambience <opinion> sucked <sentiment> NEG | It is great because sushi is tasty [SSEP] It is bad because ambience is sucked |
| I was very disappointed with the chef. | <aspect> chef <opinion> very disappointed <sentiment> NEG | It is bad because chef is very disappointed |

Table 2: Few sentences along with their corresponding targets, to fine-tune the T5 encoder-decoder framework, generated using our proposed templates vs. the ones proposed by PARAPHRASE (Zhang et al., 2021a). The target sequences highlighted in red are not semantically meaningful; especially the last example where the *customer* should be disappointed, and not the *chef*, contrary to what is meant by the paraphrased sentence.

works. Earlier ASTE approaches like JET-BERT (Xu et al., 2020), GTS-BERT (Wu et al., 2020), and Span-ASTE (Xu et al., 2021) are tagging-based approaches built upon a single encoder. Later works such as PASTE (Mukherjee et al., 2021), Unified-BART-ABSA (Yan et al., 2021), and PARAPHRASE (Zhang et al., 2021a) have experimentally demonstrated that ASTE can be better solved using seq.-to-seq. generative approaches.

Given the choice of our fine-tuning framework, we want both the encoder as well as the decoder to take advantage of our proposed pre-training strategy. Therefore, we do not obtain the aspect-aware sentiment representations from input sentences using only an encoder (although possible). Rather, we continually pre-train the full model (from its generic pre-trained checkpoint) using *aspect-based* prompts. The decoder is expected to learn from the contextualized representations of input sentences as produced by the encoder and decode aspect-aware sentiment representations for the corresponding [MASK] tokens based on the prompts it receives. Supervised contrastive learning is performed on these aspect-centric sentiment embeddings.

In order to pre-train the model, first we obtain aspect-aware sentiment representation(s) from a given sentence $sen_i$ using aspect-based prompts $p_{ij}$ for each aspect term $a_j$. It is to be noted here

that while a sentence can have multiple aspects, the combination of $(sen_i, p_{ij})$ will have a single sentiment label $s_{ij}$. We denote a data point $(sen_i, p_{ij})$ as $x_i$ and the corresponding sentiment label as $y_i$. Further, to train the model, we can now derive multiple data points from sentences with more than one aspect. This alleviates to some extent the large amount of data required to effectively (pre)train a model using contrastive learning (Li et al., 2021b).

As depicted in Fig. 1(a), we pass the input sentence through the encoder, and provide an aspect-based prompt to the decoder. The prompt consists of a [MASK] that masks the *sentiment* associated with the aspect. The decoder-generated output for the [MASK] token forms the aspect-centric sentiment representation $z_i$ corresponding to the data point $x_i$. The supervised contrastive loss on the batch $\mathcal{I}$ is defined as follows:

$$\mathcal{L}_{\mathcal{I}}^{sup} = \sum_{i \in \mathcal{I}} -\frac{1}{|P(i)|} \sum_{p \in P(i)} log \frac{exp(z_i \cdot z_p / \tau)}{\sum_{a \in A(i)} exp(z_i \cdot z_a / \tau)}$$ (1)

Here, index $i$ represents the *anchor*. $A(i) \equiv \mathcal{I} \setminus i$ represents the set of all indices except the anchor. $P(i) = \{p \in A(i) : y_p = y_i\}$ is the set of indices of all positives (same sentiment label) distinct from $i$, and $|P(i)|$ is its cardinality. The $\cdot$ symbol denotes the inner dot product between two embeddings, and $\tau \in \mathcal{R}^+$ is a scalar temperature parameter.

## 2.2 Multi-task Approach For ASTE

After being pre-trained, the encoder-decoder framework now needs to be fine-tuned for the ASTE task. For building our fine-tuning architecture, we leverage multi-task learning (MTL) which has been successfully used across a range of ABSA tasks (Collobert and Weston, 2008; He et al., 2019; Gao et al., 2022) to improve the performance of the main task by designing and training related auxiliary tasks jointly. Designing appropriate auxiliary tasks is also challenging since MTL is not guaranteed to always improve the main task performance (Martínez Alonso and Plank, 2017). Next, we describe how T5 is trained for the main task of ASTE, before elaborating on the motivation and working of two auxiliary modules as depicted in Fig. 1(b).

Corresponding to each sentence $x$ being passed as input to the encoder, first we construct the target sequence $y$ to be generated by the decoder. Let $\mathcal{T} = \{t_j \mid t_j = (asp_j, opin_j, s_j)\}_{j=1}^{|\mathcal{T}|}$ be the set of opinion triplets associated with $x$. The linearized target sequence $y$, as depicted in Fig. 1(b), takes the form as reported in Table 2. The target construction algorithm is presented in **Algorithm 1** (refer A.1). Let $e$ denote the encoder-generated contextualized representation of $x$. At the i-th time step, the decoder output $y_i = \mathcal{D}(e, y_{<i})$ is computed based on $e$ and the previous outputs $y_{<i}$. Probability distribution for the next token is obtained as:

$$p_\theta(y_{i+1}|e, y_{<i+1}) = softmax(W^T y_i) \quad (2)$$

Here, $\theta$ is initialized with parameter weights obtained after pre-training the model using contrastive learning. $W$ maps $y_i$ to a logit vector which is then used to calculate the probability distribution over the whole vocabulary set. It is to be noted here that the tokens <aspect>, <opinion>, and <sentiment> are added to the vocabulary at the time of training, and their embeddings are learnt from scratch. Finally, the model parameters are fine-tuned on the input-target pairs by minimizing the negative log-likelihood $p_\theta(y|e)$ (denoted $\mathcal{L}_{ED}$) as follows:

$$\mathcal{L}_{ED} = -log\, p_\theta(y|e) = -\sum_{i=1}^{n} log\, p_\theta(y_i|e, y_{<i}) \quad (3)$$

where $n$ is the length of the target sequence $y$.

### 2.2.1 Opinion Term Detection (OTD)

The motivation behind including this module comes from (Mrini et al., 2022) where the authors introduce a similar auxiliary module called *entity mention detection*, modeled as a token-wise binary classification task, to improve the performance of the main task of *autoregressive entity linking*, formulated as a "language generation task". We have a similar setting, where our main task of ASTE follows a generative paradigm, whereas, as depicted in Fig. 1(b), we formulate OTD as a sequence-tagging task using the BIO scheme. We hypothesize that the *opinion term detection* (OTD) module will help to better detect the *opinion span* boundaries which in turn affects the sentiment prediction. Formally, for each token $tok_i \in x$, the *opinion tagger* takes as input the contextualized token embedding generated by the encoder, and performs a 3-way classification task with the classes being **B**-beginning of the span, **I**-inside the span, **O**-outside the span. The module is trained by minimizing the *Cross Entropy* loss ($\mathcal{L}_{OTD}$) between the true and predicted labels. Our ablation results reported in Sec. 4 further justify the importance of this module.

### 2.2.2 Triplet Count Estimation (TCE)

During fine-tuning of T5 for the ASTE task, the target sequences explicitly guide the decoder, through supervision, on how many triplets to generate. In order to further augment this process, we introduce the auxiliary task of *triplet count estimation* (TCE). Part of our motivation comes from (Mrini et al., 2022) where the second auxiliary task of *entity match prediction* is modeled as a classification task. Please note however that the exact number of triplets in a test sentence is not known a priori. Hence, we design TCE as a simple *regressor*, consisting of two layers of fully connected networks (FCN). Specifically, we take the encoder-generated sentence embedding $e$ as input (768-dim.) and pass it through the first FCN layer consisting of 128 neurons. The outputs of this layer are passed through the second layer of FCN which consists of a single neuron. The regressor is trained to predict the number of triplets associated with the sentence $x$ by minimizing the *Mean Squared Error* loss ($\mathcal{L}_{TCE}$).

It is to be noted here that the TCE module, being a regressor, generates float values. For inference, we round off the output to the nearest integer (and $<integer>.5$ to $<integer>$). However, there is no direct influence of the TCE output on the T5 decoder in the absence of any explicit connection between the two. Please note that this architectural choice is consistent with the literature on multi-task learning unless the main and auxiliary modules are trained in a hierarchical setup (Sanh et al., 2019).

| Datasets | | #S | POS | NEU | NEG |
|---|---|---|---|---|---|
| | Train | 906 | 817 | 126 | 517 |
| Lap14 | Dev | 219 | 169 | 36 | 141 |
| | Test | 328 | 364 | 63 | 116 |
| | Train | 1266 | 1692 | 166 | 480 |
| 14Res | Dev | 310 | 404 | 54 | 119 |
| | Test | 492 | 773 | 66 | 155 |
| | Train | 605 | 783 | 25 | 205 |
| 15Res | Dev | 148 | 185 | 11 | 53 |
| | Test | 322 | 317 | 25 | 143 |
| | Train | 857 | 1015 | 50 | 329 |
| 16Res | Dev | 210 | 252 | 11 | 76 |
| | Test | 326 | 407 | 29 | 78 |

Table 3: ASTE-V2 dataset statistics. #S denotes the no. of sentences, 'POS', 'NEU', and 'NEG' denote the no. of positive, neutral, and negative triplets respectively.

Also, we do not post-process the decoder-generated sequence to match the same number of triplets as predicted by the TCE module before calculating the final results. The only implicit guidance between T5 and TCE is expected through the joint optimization of loss functions as described next. Our ablation results (Sec. 4) justify the advantage of TCE in improving ASTE performance.

### 2.2.3 Joint Training

Our *base* model, *CONTRASTE-Base* is trained by optimizing the encoder-decoder loss $\mathcal{L}_{ED}$ only. The full model, *CONTRASTE-MTL* is jointly trained in a multi-task setup by minimizing the combined loss $\mathcal{L}$ as follows:

$$\mathcal{L} = \mathcal{L}_{ED} + \alpha \cdot \mathcal{L}_{OTD} + \beta \cdot \mathcal{L}_{TCE} \quad (4)$$

$\alpha$ and $\beta$ are the weight coefficients assigned to the OTD loss $\mathcal{L}_{OTD}$, and TCE loss $\mathcal{L}_{TCE}$ respectively.

## 3 Experiments

### 3.1 Datasets & Evaluation Metrics

We evaluate *CONTRASTE* on four ASTE datasets (**ASTE-Data-V2**) released by Xu et al. (2020), which includes one dataset from the *laptop* domain and three datasets from the *restaurant* domain. The statistics of all the datasets are shown in Table 3.

Following prior works, we report *precision*, *recall* and *F1* scores to evaluate and compare methods on the ASTE task. A predicted triplet is considered correct if all its predicted elements exactly match with those of a ground-truth opinion triplet.

### 3.2 Experimental Setup

#### 3.2.1 Pre-Training

For this, we combine the train data from all four ASTE-DATA-V2 datasets (Table 3) to prepare our

**pre-training dataset**. This results in a total of **5,039 train** data points (refer Section 2.1) from 3,634 sentences. Please note that we do not need test data in the pre-training phase. Also, different from Li et al. (2021b), we **do not use any external data** for performing supervised contrastive pre-training. Pre-trained *t5-base*[1] was used to initialize the model weights. We (pre)train the T5 encoder-decoder framework for 14 epochs using AdamW optimizer (Loshchilov and Hutter, 2017) with a learning rate of 2e-7 and a batch size of 16. The temperature parameter $\tau$ was set to 0.07.

#### 3.2.2 Fine-Tuning

*CONTRASTE-MTL* contains around 222 million trainable parameters. For each of the datasets, we respectively fine-tune the pre-trained model weights for the downstream ASTE task using AdamW optimizer with a learning rate of 1e-4 for 14Res and 16Res, and 3e-4 for 15Res and Lap14. A batch size of 16 was used for all datasets. Following Mrini et al. (2022), we optimize the auxiliary task weights, $\alpha$ (OTD), and $\beta$ (TCE) for each dataset. As shown in Fig. 3, we start by optimizing $\alpha$ with $\beta$ set to 0.4. We then optimize $\beta$ given the optimal $\alpha$ values. One can visibly observe that the ASTE performance varies with changing task weights. For each dataset, we obtain a different set of optimal $\alpha$ and $\beta$ values based on the highest ASTE $F_1$ scores on the respective val sets. Fig 3 shows the optimal weights on all for ASTE-Data-V2 datasets; $\alpha = 1.0, \beta = 0.4$ on Lap14, $\alpha = 0.2, \beta = 0.6$ on 14Res, $\alpha = 0.8, \beta = 0.4$ on 15Res, and $\alpha = 0.8, \beta = 0.8$ on 16Res.

Each of our models was trained for 20 epochs and the model instance corresponding to the best val F1 score was used to evaluate the test set. We report the median scores over five runs of the experiments. Each pre-training epoch took 10 minutes and each fine-tuning epoch took 1 minute on *15Res* and 2 minutes on the other three datasets. All our experiments were run on Tesla P100-PCIE 16GB GPU. We make our codes publicly available.[2]

### 3.3 Baselines

We compare our proposed approach with several state-of-the-art ASTE baselines which can be broadly grouped into the following five categories:
- **Pipeline:** CMLA, RINANTE (Dai and Song, 2019) and Li-unified-R (Li et al., 2019) are

---

[1]https://huggingface.co/t5-base
[2]https://github.com/nitkannen/CONTRASTE/

| Model | 14Res | 15Res | 16Res | Lap14 |
|---|---|---|---|---|
| PARAPHRASE | 0.715 | 0.621 | 0.719 | 0.605 |
| ASTE-Base | 0.720 | 0.634 | 0.722 | 0.608 |
| w/ SCL-Sent. | 0.722 | 0.645 | 0.724 | 0.611 |
| CONTRASTE-Base | **0.728** | **0.648** | **0.730** | **0.614** |

Table 4: Test $F_1$ scores on ASTE. Comparing PARA-PHRASE with our non-multi-task model variants.

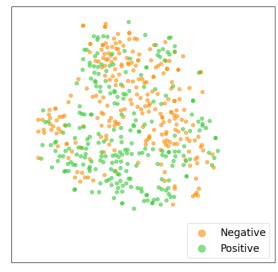 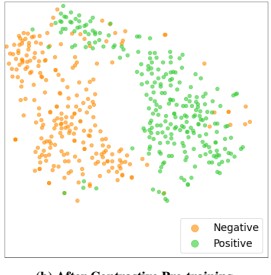

**(a) Before Contrastive Pre-training**    **(b) After Contrastive Pre-training**

Figure 2: t-SNE visualization of decoder-generated [MASK] token embeddings from aspect-based prompts derived from *15Res* val set. Our SCL objective encourages the decoder to produce discriminable representations of different sentiment polarities.

pipeline methods to co-extract aspects and opinion terms. CMLA+, RINANTE+, Li-unified-R, and Peng-two-stage are improved versions proposed by (Peng et al., 2020a) to jointly extract the aspects, opinions, and corresponding sentiments.

- **Tagging-based:** OTE-MTL (Zhang et al., 2020) propose a multi-task framework to jointly extract the three sentiment terms. JET-BERT (Xu et al., 2020) is an end-to-end approach that proposes a novel position-aware tagging scheme. GTS-BERT (Wu et al., 2020) models ASTE as a grid-tagging task. EMC-GCN (Chen et al., 2022a) proposes an Enhanced Multi-Channel GCN network to model the relation between words.
- **Span-based:** Span-ASTE (Xu et al., 2021) considers the span-level interactions of targets and opinions while predicting the sentiment. SBSK-ASTE (Feng et al., 2022) uses abundant syntax Knowledge to improve ASTE. Span-BiDir (Chen et al., 2022b) is a recent approach that uses a span-level bidirectional network to utilize all possible spans for extracting tuples bidirectionally.
- **Generative:** PASTE (Mukherjee et al., 2021) is a tagging-free decoding scheme using pointer networks for ASTE. Unified-BART-ABSA (Yan et al., 2021) unifies all ABSA tasks with a generative framework using BART. GAS (Zhang et al., 2021b) proposes a T5-based solution with linearized templated targets. PARAPHRASE (Zhang et al., 2021a) proposes to decode triplets as templated natural language paraphrases.
- **MRC-based:** BMRC (Chen et al., 2021) proposes a novel method to solve ASTE as a Bidirectional Machine Reading Comprehension task.

Among these, PARAPHRASE is closest to our fine-tuning approach, and one of the strongest baselines. Next, we demonstrate the better advantage of our prompt-based templates over the ones used in PARAPHRASE to fine-tune the model for ASTE.

### 3.4 PARAPHRASE vs. Our Templates

*CONTRASTE-Base* refers to our *base* (non-multi-task) model variant which is first pre-trained using

our proposed pre-training strategy (refer Section 2.1) and then fine-tuned for ASTE using our proposed templates (refer Table 2). We denote *ASTE-Base* as the variant of *CONTRASTE-Base* without the pre-training part. In essence *ASTE-Base* is directly comparable to PARAPHRASE (Zhang et al., 2021a), the difference being the templates used for fine-tuning. As previously discussed, the templates used in PARAPHRASE do not generalize well across ABSA tasks, and often lead to semantically meaningless or wrong targets (refer Table 2). Our prompt-based templates, on the other hand, can easily be extended across tasks, such as ACOS, as detailed in Section 4. Fine-tuning the T5 encoder-decoder framework with our templates also results in better ASTE performance as reported in Table 4. One can observe that *ASTE-Base* outperforms PARAPHRASE with overall **0.9%** $F_1$ gains across all four ASTE benchmark datasets. Detailed results for all compared models are reported in Table 5.

### 3.5 Comparison of Supervised Contrastive Pre-Training Approaches

As discussed in Section 2.1, we design novel aspect-based prompts to pre-train our encoder-decoder framework by performing supervised contrastive learning (SCL) on decoder-generated *aspect-aware* sentiment embeddings of [MASK] tokens. In order to qualitatively understand the effect of such pre-training on the aspect-based sentiment learning ability of our framework, we show in Fig. 2, a t-SNE (van der Maaten and Hinton, 2008) visualization of these embeddings on all the aspect-based prompts derived from all four ASTE-Data-V2 test sets. We observe that the *positive*, and *negative* sentiment embeddings are better clustered and more neatly separated from each other after pre-training.

| Model | 14Res | | | 15Res | | | 16Res | | | Lap14 | | |
|---|---|---|---|---|---|---|---|---|---|---|---|---|
| | P. | R. | $F_1$ | P. | R. | $F_1$ | P. | R. | $F_1$ | P. | R. | $F_1$ |
| CMLA+ ♠ | 0.392 | 0.471 | 0.428 | 0.346 | 0.398 | 0.370 | 0.413 | 0.421 | 0.417 | 0.301 | 0.369 | 0.332 |
| RINANTE+ ♠ | 0.314 | 0.394 | 0.350 | 0.299 | 0.301 | 0.300 | 0.257 | 0.223 | 0.239 | 0.217 | 0.187 | 0.201 |
| Li-unified-R ♠ | 0.410 | 0.674 | 0.510 | 0.447 | 0.514 | 0.478 | 0.373 | 0.545 | 0.443 | 0.406 | 0.443 | 0.423 |
| Peng-two-stage ♠ | 0.432 | 0.637 | 0.515 | 0.481 | 0.575 | 0.523 | 0.470 | 0.642 | 0.542 | 0.374 | 0.504 | 0.429 |
| ChatGPT ♣ | 0.513 | 0.629 | 0.565 | 0.419 | 0.606 | 0.495 | 0.449 | 0.617 | 0.520 | 0.351 | 0.489 | 0.409 |
| OTE-MTL | 0.630 | 0.551 | 0.587 | 0.579 | 0.427 | 0.489 | 0.603 | 0.534 | 0.565 | 0.492 | 0.405 | 0.451 |
| JET-BERT ♠ | 0.706 | 0.559 | 0.624 | 0.645 | 0.520 | 0.575 | 0.704 | 0.584 | 0.638 | 0.554 | 0.473 | 0.510 |
| PASTE | 0.648 | 0.638 | 0.643 | 0.583 | 0.567 | 0.575 | 0.655 | 0.644 | 0.650 | 0.550 | 0.516 | 0.532 |
| GTS-BERT | 0.680 | 0.676 | 0.678 | 0.627 | 0.555 | 0.589 | 0.654 | 0.680 | 0.667 | 0.561 | 0.530 | 0.545 |
| GAS | 0.650 | 0.695 | 0.672 | 0.561 | 0.618 | 0.588 | 0.661 | 0.687 | 0.674 | 0.571 | 0.540 | 0.555 |
| Unified-BART-ABSA | 0.655 | 0.650 | 0.653 | 0.591 | 0.594 | 0.593 | 0.666 | 0.687 | 0.676 | 0.614 | 0.562 | 0.587 |
| BMRC | 0.756 | 0.618 | 0.680 | 0.685 | 0.534 | 0.601 | 0.712 | 0.611 | 0.658 | 0.706 | 0.490 | 0.578 |
| EMC-GCN | 0.712 | 0.724 | 0.718 | 0.615 | 0.625 | 0.619 | 0.656 | 0.713 | 0.683 | 0.617 | 0.563 | 0.588 |
| Span-ASTE | 0.729 | 0.709 | 0.719 | 0.622 | 0.645 | 0.633 | 0.695 | 0.712 | 0.703 | 0.634 | 0.558 | 0.594 |
| PARAPHRASE | 0.711 | 0.719 | 0.715 | 0.604 | 0.639 | 0.621 | 0.701 | 0.739 | 0.719 | 0.634 | 0.578 | 0.604 |
| SBSK-ASTE | 0.746 | 0.715 | 0.730 | 0.653 | 0.637 | 0.645 | 0.708 | 0.720 | 0.714 | 0.656 | 0.565 | 0.607 |
| Span-BiDir | 0.764 | 0.724 | **0.743** | 0.699 | 0.604 | 0.648 | 0.716 | 0.726 | 0.721 | 0.657 | 0.599 | 0.627 |
| ASTE-Base | 0.718 | 0.721 | 0.720 | 0.616 | 0.654 | 0.634 | 0.695 | 0.751 | 0.722 | 0.635 | 0.584 | 0.608 |
| CONTRASTE-Base | 0.724 | 0.732 | 0.728 | 0.626 | 0.672 | 0.648 | 0.721 | 0.739 | 0.730 | 0.639 | 0.591 | 0.614 |
| CONTRASTE-MTL | 0.736 | 0.744 | 0.740 | 0.653 | 0.667 | **0.661** | 0.722 | 0.763 | **0.742** | 0.642 | 0.617 | **0.629** |

Table 5: Comparative results on the ASTE-Data-V2 (Xu et al., 2020). ♠ denotes that the results are retrieved from Xu et al. (2020). ♣ ChatGPT results are obtained using 100-shot In Context Learning (ICL) prompts. The results for all other methods were reproduced using released codes and original parameters. The highest F1 scores on each dataset are highlighted in **bold**. The second highest F1 scores are underlined.

Thus, our SCL-based pre-training objective helps in improving the performance on ABSA tasks.

Next, we wanted to compare our pre-training strategy with the one used in existing ABSA works. While Li et al. (2021b) uses SCL for pre-training, Liang et al. (2021), and Ke et al. (2021) use it for fine-tuning their respective models. However, different from us, all of them apply SCL on *sentence-level* sentiment representations of sentences. In order to replicate their methodology, we pre-train our encoder-decoder framework by applying SCL on mean-pooled representations of sentences from the final layer of encoder. For this, we collected a total of **3358 data points** (sentences containing triplets with the same sentiment polarity) from 3,634 sentences combining all four train datasets. Model weights were initialized with pre-trained *t5-base*, and the framework was (pre)trained for 14 epochs using AdamW optimizer with a learning rate of 2e-5, batch size of 16, and $\tau = 0.07$.

We fine-tune *ASTE-base* from this pre-trained checkpoint (settings discussed in Section 3.2) respectively for each of the datasets and report our results in Table 4 (row ASTE-Base w/ SCL-Sent.). We observe that CONTRASTE-Base outperforms ASTE-Base w/ SCL-Sent. with overall **0.6%** improvement in F$_1$ scores. This establishes the better

suitability of performing supervised contrastive pre-training on *aspect-centric* sentiment embeddings, since in ABSA, the sentiments are defined at an aspect level and not at the sentence level.

### 3.6 Main Results

We report the comparison of our model variants with all considered baselines in Table 5. The majority of the baselines including OTE-MTL, JET-BERT, PASTE, GTS-BERT, BMRC, Span-ASTE, EMC-GCN, SBSK-ASTE, and Span-BiDir use BERT (Devlin et al., 2019) as their backbone. Unified-BART-ABSA uses BART (Lewis et al., 2020). GAS and PARAPHRASE are based on a T5 encoder-decoder framework and hence are most similar to our approach. PARAPHRASE is our strongest T5-based baseline. SBSK-ASTE and Span-BiDir are very recent span-based techniques and they qualify to be our strongest two baselines.

We find that ASTE-Base, our non-multi-task non-pre-trained model outperforms PARA-PHRASE with overall 0.9% F$_1$ gains, however, fails to beat SBSK-ASTE and Span-BiDir. This observation may be attributed to better feature learning strategies employed by these two methods. We find that our proposed SCL-based pre-training approach hugely improves the performance of ASTE-Base, as CONTRASTE-Base out-

| CON | OTD | TCE | Triplet | Aspect | Opinion | Sentiment |
|-----|-----|-----|---------|--------|---------|-----------|
| ✗ | ✗ | ✗ | 0.671 | 0.820 | 0.815 | 0.753 |
| ✓ | ✗ | ✗ | 0.680 | 0.824 | 0.820 | 0.765 |
| ✗ | ✓ | ✓ | 0.678 | 0.827 | 0.827 | 0.762 |
| ✓ | ✗ | ✓ | 0.685 | 0.832 | 0.828 | 0.770 |
| ✓ | ✓ | ✗ | 0.682 | 0.836 | 0.842 | 0.776 |
| ✓ | ✓ | ✓ | 0.692 | 0.840 | 0.848 | 0.784 |

Table 6: Ablation Results. We report F1 scores for *Triplet*, *Aspect*, and *Opinion* predictions, and *Sentiment* accuracies for correctly predicted tuples (asp, opin).

| Model | Res15 | Res16 | Rest-ACOS | Lap-ACOS |
|-------|-------|-------|-----------|----------|
| ChatGPT | 0.262 | 0.386 | 0.391 | 0.237 |
| PARAPHRASE | 0.468 | 0.578 | 0.592 | 0.429 |
| ACOS-Base | 0.458 | 0.583 | 0.597 | 0.431 |
| ACOS-Contra | **0.478** | **0.598** | **0.605** | **0.446** |

Table 7: Test $F_1$ scores on ACOS. Comparing ChatGPT and PARAPHRASE with our ACOS model variants.

performs SBSK-ASTE with overall 1.4% $F_1$ gains, while performing comparably with Span-BiDir. Finally, CONTRASTE-MTL comfortably outperforms PARAPHRASE with overall **4.4%** $F_1$ gains, SBSK-ASTE with overall **2.8%** $F_1$ gains, and Span-BiDir with overall **1.2%** $F_1$ gains, to achieve **new state-of-the-art ASTE results**.

## 4 Analysis

### 4.1 Experiments With ChatGPT

Natural Language Processing, in recent times, has been revolutionized by the evolution of Large Language Models (LLMs) such as GPT-3 (Brown et al., 2020). ChatGPT (OpenAI, 2023b), powered by GPT-3.5 and GPT-4 (OpenAI, 2023a), has pioneered the excitement around LLMs by achieving remarkable zero-shot and few-shot in-context learning (ICL) (Brown et al., 2020) results for unseen NLP tasks, without any parameter updates.

In this work, we investigated how well can ChatGPT perform on ABSA tasks. We experimented with 4 different kinds of zero-shot and few-shot prompts as detailed in Sec. A.2. For each task, our best test set results are obtained when we prompt ChatGPT with task-specific instructions followed by 100 randomly selected (sentence, target output) samples from the corresponding training set. ASTE results obtained with ChatGPT are reported in Table 5. Compared to our *CONTRASTE-MTL* results, we observe a huge gap in performance. This demonstrates that ChatGPT is far from producing SOTA ASTE results (refer Sec. A.2 for details).

### 4.2 Model Ablations

Contrastive pre-training (CON), OTD, and TCE are the three crucial components of CONTRASTE-MTL. Here, we ablate one component at a time and report in Table 6 the F1 scores corresponding to the triplet, aspect term, and opinion term predictions averaged over all four datasets. Sentiment prediction accuracies corresponding to correctly predicted opinion tuples (aspect term, opinion term) are also reported. The first row corresponds to ASTE-Base, and the last row corresponds to CONTRASTE-MTL results from Table 5.

From Table 6, we observe (bottom to top) that removing TCE and OTD degrades the triplet F1 scores by 1.3%, and 0.8% respectively, thereby highlighting their impact on the overall task. Removing TCE results in more spurious triplets being generated, which in turn hampers the scores. Removing OTD especially affects the opinion term extraction (F1 ↓ 2.4%) and sentiment prediction (acc ↓ 1.8%) performance. Removing CON substantially affects all the scores (triplet, aspect, opinion F1 scores ↓ 1.9%, 1.6%, 2.5% respectively, sentiment acc ↓ 2.9%), thereby demonstrating the advantage of contrastive pre-training using our proposed approach. This observation is further strengthened when we compare the second row, corresponding to CONTRASTE-Base, with the first row, corresponding to ASTE-Base, and see a huge jump in sentiment acc (↑ 1.6%). Finally, removing all components results in a significant drop across all the scores, thereby establishing the importance of all proposed components of CONTRASTE-MTL.

### 4.3 Qualitative Analysis

Table 8 compares the model predictions of PARAPHRASE and CONTRASTE-MTL for a few test sentences. The first example highlights the advantage of both TCE and OTD as we correctly predict both the number of triplets, as well as the triplets themselves (and hence the opinion terms) whereas PARAPHRASE misses out on both. The second example further establishes the importance of OTD as we correctly predict the opinion term boundaries whereas PARAPHRASE fails to do so.

### 4.4 Advantage of Pre-Training On ACOS

We consider the task of Aspect Category Opinion Sentiment (ACOS) quad prediction (Cai et al., 2021; Zhang et al., 2021a) in order to understand whether our proposed SCL-based pre-training tech-

| SENTENCE | TRIPLETS |
|---|---|
| Gorgeous place ideal for a romantic dinner. | GOLD: place ; Gorgeous ; POS \| place ; ideal ; POS
PARAPHRASE: place ; romantic ; POS
CONTRASTE-MTL: place ; Gorgeous ; POS \| place ; ideal ; POS |
| I complained to the manager, but he was not even apologetic. | GOLD: manager ; not even apologetic ; NEG
PARAPHRASE: manager ; apologetic ; NEG
CONTRASTE-MTL: manager ; not even apologetic ; NEG |

Table 8: Few test set sentences; their gold triplets, & predictions made by PARAPHRASE and CONTRASTE-MTL.

nique can improve the downstream performance of more difficult ABSA tasks as well. Here, given a sentence, for example, "the spicy tuna roll was unusually good", the task is to extract the (aspect, category, opinion, sentiment) quad ('spicy tuna roll', 'food quality', 'unusually good', POS). Also, there could be multiple such quads in a given sentence.

We perform our experiments on four datasets, *Restaurant-ACOS*, *Laptop-ACOS* released by Cai et al. (2021), and *Rest15*, *Rest16* released by the authors of PARAPHRASE (Zhang et al., 2021a), and report our results in Table 7. Experimental details can be found in Sec. A.3. Here, *ACOS-Base* refers to T5 trained with our proposed fine-tuning templates (updated for ACOS). *ACOS-Contra* refers to the fine-tuning of ACOS-Base from a checkpoint pre-trained using our proposed SCL-based strategy (refer Sec. 2.1). We observe that while the performance of ACOS-Base is comparable to that of PARAPHRASE, supervised contrastive pre-training of the model hugely improves the scores as ACOS-Contra outperforms both PARAPHRASE, and ACOS-Base with overall **2.9% $F_1$** gains. Also, ACOS-Contra substantially outperforms ChatGPT.

Baselines and comparative results for Target Aspect Sentiment Detection (TASD), and Aspect Extraction and Sentiment Classification (AESC) are respectively reported in Sections A.4 and A.5.

## 5   Related Works

Supervised Contrastive Learning (SCL) has been explored previously in the ABSA domain with very different objectives and approaches than ours. Li et al. (2021b) use supervised contrastive pre-training to effectively learn implicit sentiments. However, their approach is not optimal as they apply contrastive learning on sentence-level sentiment representations of sentences, whereas in ASTE (or any other ABSA task), sentiments are defined at an aspect-level. Additionally, they rely on external large-scale sentiment-annotated corpora to learn sentiment knowledge. We, differ from them as we do not use any external data. Also, we per-

form CL on aspect-aware sentiment representations of masked *opinion* terms expressing the sentiment.

Liang et al. (2021) try to distinguish between aspect-invariant and aspect-dependent sentiment features. Further, their data augmentation strategy masks the *aspect* terms, whereas we mask the *sentiment*. Finally, they apply contrastive learning only to fine-tune the BERT encoder, whereas we use SCL to pre-train an encoder-decoder setup.

Ke et al. (2021) apply contrastive learning in a continual learning setup with very different objectives of knowledge transfer across tasks, and knowledge distillation from old tasks to the new task. Moreover, all these works tackle a relatively easier downstream ABSA task of Aspect Sentiment Classification (ASC), where the goal is to determine the sentiment, given the sentence and a specific aspect. Also, all of them are encoder-based approaches. The novelty of our approach, therefore lies in designing **aspect-based** prompts, obtaining **aspect-aware** sentiment representations of masked *sentiment* terms, and applying contrastive learning on these embeddings to (pre)train an **encoder-decoder** framework for the downstream ASTE task.

## 6   Conclusion

We propose a novel strategy to continually pre-train T5, from its publicly available generic pre-trained checkpoint, for ABSA tasks. For this, given a sentence, first, we derive aspect-based prompts with corresponding sentiments *masked*. We then apply supervised contrastive learning (SCL) on the decoder-generated *aspect-level* sentiment embeddings of the masked tokens. Compared to performing SCL on *sentence-level* sentiment embeddings, we show that our strategy results in a better downstream performance for ABSA tasks such as ASTE, ACOS, TASD, and AESC. Also, we do not use any external data for pre-training. For task-specific fine-tuning, we propose generic placeholder-based templates to train T5. Finally, we present our multi-task model, *CONTRASTE-MTL*, that achieves SOTA results on the majority of ASTE benchmark datasets.

## 7 Limitations

It requires a substantial amount of data to effectively apply contrastive learning (Li et al., 2021b) even for continually pre-training an already pre-trained (generic) encoder-decoder model such as T5. However, in this work, we limited ourselves to the training sets of existing ASTE benchmarks. Using our proposed aspect-based prompts, we could obtain more data points than the actual number of sentences, and they were sufficient enough for us to achieve new state-of-the-art results. The impact of pre-training with more data however remains to be investigated.

While preparing the pre-training data, we combined data from both domains, that is, *laptop* and *restaurant*. It was done mainly because our experiments with pre-training the model with a limited number of *laptop* data points could not give us good results (please note that there are three restaurant datasets vs one laptop dataset in the ASTE-Data-V2 benchmark). This raises an important question regarding how much data is sufficient enough to effectively pre-train the model. Running pre-training experiments with different fractions of available data is something that we have not explored here. Also, the applicability of our proposed scheme in cross-domain settings remains to be investigated.

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

| Model | 14Res | | | 15Res | | | 16Res | | | Lap14 | | |
|---|---|---|---|---|---|---|---|---|---|---|---|---|
| | P. | R. | F$_1$ | P. | R. | F$_1$ | P. | R. | F$_1$ | P. | R. | F$_1$ |
| 5-shot ICL (Han et al., 2023) | - | - | 0.549 | - | - | 0.466 | - | - | 0.518 | - | - | 0.390 |
| Setting 1 | 0.151 | 0.175 | 0.162 | 0.125 | 0.181 | 0.148 | 0.184 | 0.255 | 0.214 | 0.171 | 0.233 | 0.197 |
| Setting 2 | 0.445 | 0.529 | 0.484 | 0.377 | 0.551 | 0.447 | 0.426 | 0.609 | 0.502 | 0.276 | 0.375 | 0.318 |
| Setting 3 | 0.513 | 0.629 | 0.565 | 0.419 | 0.606 | 0.495 | 0.449 | 0.617 | 0.520 | 0.351 | 0.489 | 0.409 |
| Setting 4 | 0.488 | 0.619 | 0.546 | 0.400 | 0.575 | 0.472 | 0.473 | 0.656 | 0.549 | 0.351 | 0.475 | 0.404 |
| CONTRASTE-MTL | 0.736 | 0.744 | **0.740** | 0.653 | 0.667 | **0.661** | 0.722 | 0.763 | **0.742** | 0.642 | 0.617 | **0.629** |

Table 9: Comparing ASTE results obtained using ChatGPT and CONTRASTE-MTL. All four settings are described in the text. Setting 3 gives us our best ChatGPT results, however substantially outperformed by CONTRASTE-MTL.

# A  Appendix

## A.1  Algorithm For Target Construction

---

**Algorithm 1** Converting opinion triplets associated with a sentence into a linearized target sequence

---

$Input : triplets$: [(aspect, opinion, sentiment), ...] - List of opinion triplets for the given sentence.
$linear\_string$ = ""
**for** $triplet$ in $triplets$ **do**
    $linear\_string$ += <aspect>
    $linear\_string$ += $triplet[0]$
    $linear\_string$ += <opinion>
    $linear\_string$ += $triplet[1]$
    $linear\_string$ += <sentiment>
    $linear\_string$ += $triplet[2]$
    $linear\_string$ += [SSEP]
**end for**
$linear\_string$ = $linear\_string[:-6]$

---

## A.2  Experiments With ChatGPT

Natural Language Processing, in recent times, has been revolutionized by the evolution of Large Language Models (LLMs) such as GPT-3 (Brown et al., 2020), PaLM (Chowdhery et al., 2022), Llama 2 (Touvron et al., 2023), etc. ChatGPT (OpenAI, 2023b), powered by GPT-3.5 and GPT-4 (OpenAI, 2023a), has pioneered the excitement around LLMs by achieving remarkable zero-shot and few-shot in-context learning (ICL) (Brown et al., 2020) results for unseen tasks, without any parameter updates.

In this work, we carried out exhaustive experiments to investigate how well can ChatGPT solve ABSA tasks. The ChatGPT version used by us is *gpt-3.5-turbo*. The *temperature* parameter was set to 0 in order to avoid variations in ChatGPT-generated outputs. This will help us to reproduce the results in the future with the same settings. For generating the responses, the *max_tokens* parameter was set to 512. We experimented with 4 different prompts as detailed below to get the response for each test set sentence. We take the example of ASTE to explain our experimental settings:

- **Setting 1: Full zero-shot** with no task defini-

tion. Here the prompt that we use is: Given the restaurant/laptop review: sent, find all the (aspect, opinion, sentiment) triplets appearing in the review in JSON format, with review and triplets as the keys. No explanations are needed and do not provide any text in the response.

- **Setting 2: Zero-shot with task definition**. Here, we prepend the following ASTE task definition to the prompt mentioned above: Aspects are nouns or phrases appearing in the text that indicate specific attributes of the entity being reviewed. Opinions are adjectives or phrases that express specific sentiments towards the aspects. Sentiments could be either positive, negative, or neutral.

- **Setting 3: Few-shot In-Context-Learning (ICL) with the same exemplars**. To design the prompt in this setting, after the task definition mentioned in Setting 2, and before the final instruction defined in Setting 1 above, we add 100 randomly selected (sentence, target output) samples from the corresponding training set. Please note that the same 100 samples are used as part of the prompt for each test set sentence.

- **Setting 4: Few-shot ICL with dynamically selected exemplars**. Here, for each test set sentence, we use a different set of randomly selected 100 samples from the corresponding training set as part of the prompt. While no model parameters are updated in our experiments with ChatGPT, our goal in this setting is to investigate whether covering a broader range of training set samples helps to improve scores.

ASTE results obtained using ChatGPT are reported and compared against CONTRASTE-MTL in Table 9 above. First, we observe that across all 4 datasets, using the task definition helps to improve scores considerably. Prompting with training set examples further improves the scores. However, we do not observe any advantage of Setting 4 over Setting 3. This is expected since the model does not remember the training set examples it

| Model | Res15 | | | Res16 | | | Rest-ACOS | | | Lap-ACOS | | |
|---|---|---|---|---|---|---|---|---|---|---|---|---|
| | P. | R. | F₁ | P. | R. | F₁ | P. | R. | F₁ | P. | R. | F₁ |
| ChatGPT | 0.257 | 0.271 | 0.262 | 0.367 | 0.407 | 0.386 | 0.396 | 0.386 | 0.391 | 0.253 | 0.224 | 0.237 |
| Paraphrase | 0.457 | 0.479 | 0.468 | 0.575 | 0.582 | 0.578 | 0.597 | 0.588 | 0.592 | 0.434 | 0.424 | 0.429 |
| ACOS-Base | 0.451 | 0.465 | 0.458 | 0.569 | 0.598 | 0.583 | 0.598 | 0.597 | 0.597 | 0.436 | 0.426 | 0.431 |
| ACOS-Contra | 0.471 | 0.484 | **0.478** | 0.587 | 0.610 | **0.598** | 0.608 | 0.602 | **0.605** | 0.450 | 0.441 | **0.446** |

Table 10: Comparative ACOS results on the Lap-ACOS, Rest-ACOS (Cai et al., 2021), and Res15, Res16 (Zhang et al., 2021a) datasets. ChatGPT results are obtained using 100-shot In Context Learning (ICL) prompts. The highest F1 scores on each dataset are highlighted in **bold**. The second highest F1 scores are underlined.

was prompted with earlier while generating the response for a new test set sentence. Its memory is limited to the context of the prompt.

We also compare our ChatGPT results with the ones reported in (Han et al., 2023) where the authors investigate the capability of ChatGPT in solving different information extraction tasks, including ABSA. We observe that our Setting 3/4 results comfortably outperform them. Finally, comparing our Setting 3 results with the ones obtained using our proposed CONTRASTE-MTL model, we observe a huge gap in performance. With necessary task-specific adjustments made to Setting 3, we report the ChatGPT results for other ABSA tasks in tables 10, 11, and 12 respectively, and observe similar trends in performances. These observations reinstate the findings of Han et al. (2023) and demonstrate that ChatGPT is not equipped enough to produce state-of-the-art results for ASBA tasks.

### A.3 Advantage of Pre-Training On ACOS

We perform our experiments on four benchmark ACOS datasets, *Restaurant-ACOS*, *Laptop-ACOS* released by Cai et al. (2021), and *Rest15*, *Rest16* released by the authors of PARAPHRASE (Zhang et al., 2021a), and report our **detailed results** in Table 10. We had to make a minor change in our fine-tuning templates by adding the special token `<category>` and its associated value **before** `<aspect>`. The remaining fine-tuning settings are similar to ASTE as discussed in Sec. 3.2.

### A.4 Advantage of Pre-Training On TASD

Here, we consider the task of Target (category) Aspect Sentiment triplet Detection (TASD) (Wan et al., 2020). Given a sentence, for example, "the spicy tuna roll was unusually good", the task is to extract the (category, aspect, sentiment) triplet ('food quality', 'spicy tuna roll', POS). Further, there could be multiple such triplets in a sentence. We perform our experiments on two benchmark

| Model | Res15 | Res16 |
|---|---|---|
| ChatGPT | 0.469 | 0.561 |
| TAS-T5-SW-BIO-CRF | 0.533 | 0.616 |
| BART-Phrase-Joint-TASD | 0.585 | 0.602 |
| T5-Phrase-Joint-TASD | 0.614 | 0.698 |
| T5-Sentence-Joint-TASD | 0.611 | 0.675 |
| GAS-Extraction | 0.615 | 0.694 |
| PARAPHRASE | 0.630 | 0.719 |
| LEGO-ABSA | 0.617 | 0.688 |
| Seq2Path (k=8) | 0.633 | 0.721 |
| EHG-Para | 0.628 | 0.721 |
| TASD-Base | 0.634 | 0.709 |
| TASD-Contra | **0.664** | **0.747** |

Table 11: Test F₁ TASD scores on the benchmark Res15, and Res16 datasets (Wan et al., 2020). The highest F1 scores on each dataset are highlighted in **bold**. The second highest F1 scores are underlined.

datasets, *Res15*, and *Res16* released by Wan et al. (2020), and report our results in Table 11.

Among the baselines, TAS (Wan et al., 2020) highlights that the prediction of sentiment polarities depends on both the target (category) as well as the aspect terms, and accordingly proposes a novel approach for target-aspect-sentiment joint detection, specifically improving the methods that predict sentiment polarities based only on explicit targets. Joint-TASD (Chebolu et al., 2021) transforms ABSA into abstract summary-like conditional text generation. GAS-Extraction (Zhang et al., 2021b) is a predecessor of the PARAPHRASE paper that generates triplets with simple linearized tuple templates. PARAPHRASE (Zhang et al., 2021a) as discussed before proposes to model triplet generation as natural language paraphrases. LEGO-ABSA (Gao et al., 2022) solves multiple ABSA tasks by controlling the task prompts. Seq2Path (Mao et al., 2022) proposes a novel method where they generate sentiment tuples as paths of a tree. EHG-Para (Lv et al., 2023) leverages a novel Efficient Hybrid

| Model | 14Res | 15Res | 16Res | Lap14 |
|---|---|---|---|---|
| ChatGPT | 0.634 | 0.556 | 0.623 | 0.541 |
| Unified-BART-ABSA | 0.785 | 0.699 | 0.757 | 0.682 |
| GAS-R | 0.790 | 0.688 | 0.757 | 0.658 |
| EHG | 0.793 | 0.700 | 0.771 | 0.685 |
| SentiPrompt | 0.811 | **0.742** | 0.798 | 0.708 |
| AESC-Base | 0.818 | 0.729 | 0.781 | 0.701 |
| AESC-Contra | **0.826** | 0.741 | **0.810** | **0.731** |

Table 12: Comparative test set AESC results on the benchmark datasets (Peng et al., 2020b). The highest and the second highest F1 scores on each dataset are respectively highlighted in **bold** and underline.

Transformer and proposes a novel global hybrid loss in combination with bipartite matching.

In Table 11, *TASD-Base* refers to the T5 encoder-decoder framework trained with our proposed fine-tuning templates (updated for TASD). *TASD-Contra* refers to the fine-tuning of TASD-Base from a checkpoint that is pre-trained using our proposed strategy (refer Section 2.1). We observe that TASD-Base performs slightly better on Res15 than the three strongest baselines, EHG-Para, PARAPHRASE, and Seq2Path. However, its performance on Res16 is weaker than the other three. Leveraging the advantages of supervised contrastive pre-training, TASD-Contra performs substantially better than TASD-Base while comfortably outperforming EHG-Para, PARAPHRASE, and Seq2Path with overall 4.67%, 4.65%, and 1.88% $F_1$ gains, averaged over the two datasets. For our experiments, here again we had to make a minor change in our fine-tuning templates by adding the special token `<category>` and its associated value before `<aspect>`. Also, we remove the `<opinion>` placeholders from the targets.

### A.5 Advantage of Pre-Training On AESC

Lastly, we consider the task of Aspect Extraction and Sentiment Classification (AESC). Here, given a sentence, for example, "the spicy tuna roll was unusually good", the task is to extract the (aspect, sentiment) pair ('spicy tuna roll', POS). Further, there could be multiple such pairs in a given sentence. We perform our experiments on a total of four benchmark datasets (Peng et al., 2020b) from two domains; 14Res, 15Res, and 16Res belonging to the *Restaurant* domain, and Lap14 built on *Laptop* reviews, and report our results in Table 12.

Among the baselines, Unified-BART-ABSA (Yan et al., 2021) converts all ABSA subtasks into a unified generative formulation by redefining respec-

tive subtask targets as sequences mixed with aspect term/opinion term word indices and sentiment class indexes. GAS-R (Zhang et al., 2021b) again tackles ABSA tasks in a unified generative framework by formulating targets using simple linearized templates. EHG (Lv et al., 2023) leverages a novel Efficient Hybrid Transformer to generate the semantic and position information of AESC targets in parallel. SentiPrompt (Li et al., 2021a) injects sentiment knowledge connecting aspects and opinion terms using sentiment knowledge-enhanced prompts to tune the language model.

Similar to the previously discussed tasks, in Table 12, *AESC-Base* and *AESC-Contra* refer to our fine-tuned T5 model variants (with templates suitably updated for AESC) without and with contrastive pre-training respectively. Consistent with our previous observations, here again we establish the advantages of our proposed pre-training strategy as AESC-Contra outperforms AESC-Base and our strongest competitor SentiPrompt with overall 2.65%, and 1.65% $F_1$ gains, averaged across the four datasets. For our experiments, our fine-tuning templates only consist of the `<aspect>` and `<sentiment>` placeholders followed by their respective values. All the above experiments further highlight the generalizability of our templates over PARAPHRASE. When performing our experiments for the respective tasks, i.e. ACOS, TASD, and AESC, we did not make any changes in the pre-training settings. The fine-tuning settings are similar to ASTE as discussed in Section 3.2.

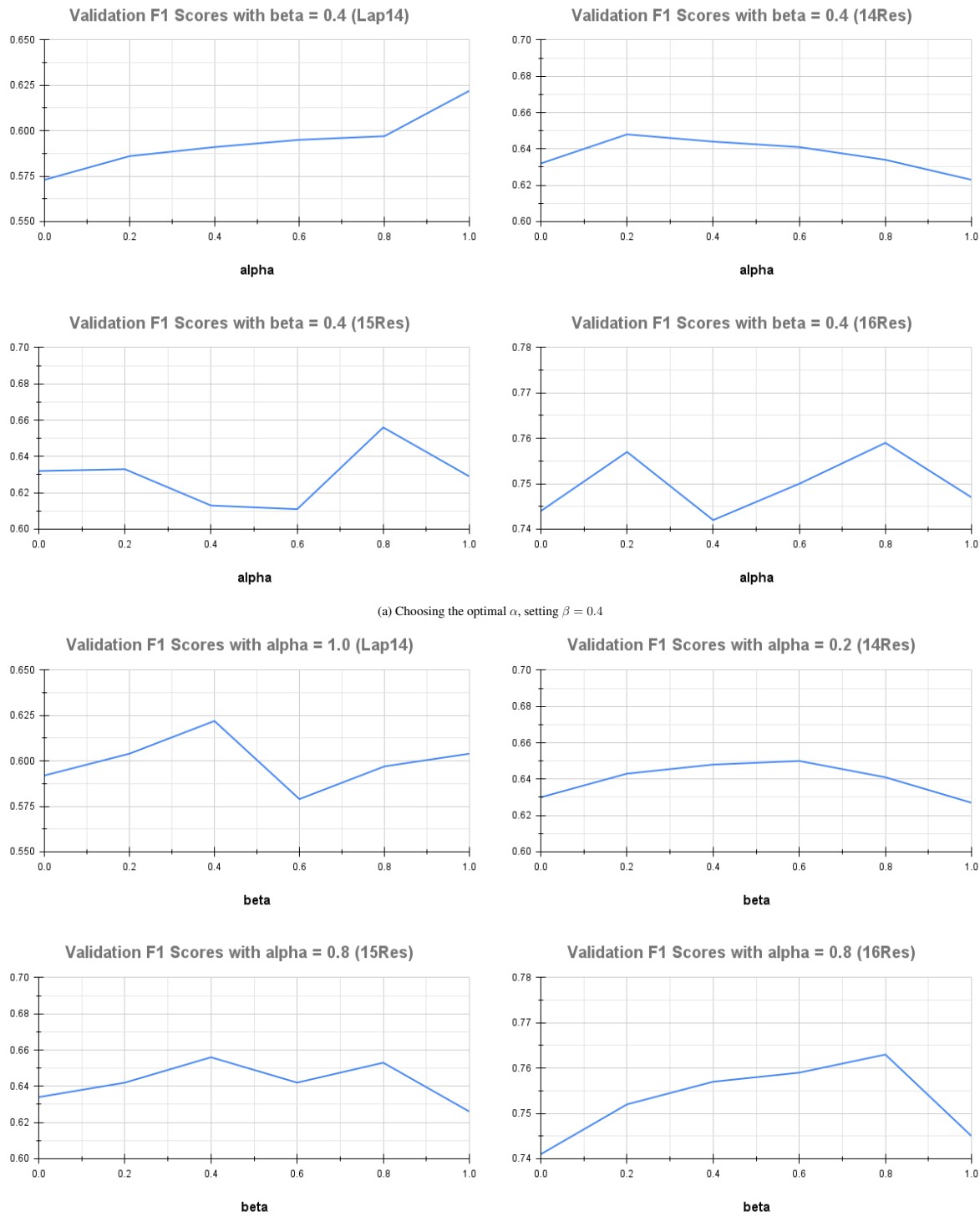

(a) Choosing the optimal $\alpha$, setting $\beta = 0.4$

(b) Choosing the optimal $\beta$, given the optimal $\alpha$

Figure 3: Task weight tuning on the dev set for Opinion Term Detection (OTD) and Triplet Count Estimation (TCE). We first optimize for $\alpha$ (a), and then for $\beta$ (b).