# OpenReview forum: "CONTRASTE: Supervised Contrastive Pre-training With Aspect-based Prompts For Aspect Sentiment Triplet Extraction"
_EMNLP/2023/Conference — EMNLP 2023 Findings_

### Official Review · Reviewer_RHnr · 2023-07-28

**Paper Topic And Main Contributions:** 1、In this work, the authors focus on …
**Soundness:** 4

**Excitement:**

4: Strong: This paper deepens the understanding of some phenomenon or lowers the barriers to an existing research direction.

**Reasons To Accept:**

1、The authors provide a clear statement of the research problem, and the overall content demonstrates a relatively coherent logic.
2、The methods employed in the paper, including the reasonable adoption of pre-training techniques and the incorporation of OTD and TCE modules during fine-tuning, are well-justified. Additionally, the explanations of the methods are presented in a clear manner.
3、The authors conducted extensive and comprehensive experiments to support their methods, and they also performed supplementary experiments on other ABSA tasks.

**Reasons To Reject:**

1、	In the comparative experiments with the relevant methods and Baseline, it is recommended to include additional novel approaches.
2、	The description of the TCE module in the paper's methodology is concise, while further elaboration is required on the specific procedural steps for utilizing the TCE module to guide the generation of triplet quantities by the model.

**Reproducibility:**

4: Could mostly reproduce the results, but there may be some variation because of sample variance or minor variations in their interpretation of the protocol or method.

**Reviewer Confidence:**

4: Quite sure. I tried to check the important points carefully. It's unlikely, though conceivable, that I missed something that should affect my ratings.

---

> ### Author Rebuttal · Authors · 2023-08-29
>
> Thank you for your time and effort in giving us your valuable feedback. We are extremely happy to note that you have generously appreciated our coherent presentation, the motivation and novelty of our proposed pre-training strategy, and its utility in achieving state-of-the-art results for multiple ABSA tasks, the justification behind the inclusion of OTD and TCE modules for fine-tuning, and finally the comprehensiveness of our experiments. Please find below our clarifications/answers to your queries:
>
> 1. "...recommended to include additional novel approaches."
>
> From Table 5, please note that we have compared our ASTE results with a total of 16 different baseline approaches, which, to the best of our knowledge, is the most by any published paper on the task. Further, in order to demonstrate the novelty of our proposed approach, and the importance of our design decisions, we report the scores for 3 different variants of our model, ASTE-Base, CONTRASTE-Base, and CONTRASTE-MTL. As per the suggestions of other reviewers, we have also performed detailed experiments with chatGPT, which we plan to include in the final version. We try different zero-shot and few-shot prompts to find out that CONTRASTE-MTL results are substantially better than the best results obtained using ChatGPT. We observe the same trend for all other ABSA tasks. We therefore conclude that ChatGPT is not equipped enough to give us state-of-the-art results for ASBA tasks. Here, we would request you to kindly provide us with some references for papers which we might have missed unknowingly. We assure you to include them as part of our camera-ready draft.
>
> 2. "The description of the TCE module in the paper's methodology is concise.."
>
> Thank you for pointing this out. As mentioned on line 288, the Triplet Count Estimator (TCE) is a simple regressor built on top of the encoder using fully connected networks (FCN). More specifically, we take the mean encoder embeddings of the input sentence (dim 768) and pass it to the first FCN (dim 128 X 768). Next, we pass the outputs of this layer to the second FCN (dim 1 X 128) that brings it down to a float value. The output of the layer is passed into the MSE loss function along with the ground truth no. of triplets. As mentioned on line 301, the MSE loss is jointly minimized with a weighing term along with the other 2 objectives.
>
> Also, please note that the number of triplets varies across sentences. Also, a test set sentence may well contain more triplets than any other sentence it has seen during training. Hence, we modeled the auxiliary subtask of triplet count estimation (TCE) as a regressor instead of a classifier since we do not know the exact no. of triplets apriori. Please note from line 287, that TCE is introduced to implicitly guide the decoder when to stop. As mentioned before, this implicit guidance is expected through the joint optimization of loss functions for the three modules, encoder-decoder, OTD and TCE. Please note that there is no explicit connection between the output of the TCE module and the decoder. As mentioned on line 283, the explicit guidance for the decoder on how many triplets to generate is provided through supervision from ground truth target sequences.
>
> We assure you to include these details in the camera-ready draft. Also, we would be more than happy to answer any follow-up questions that you may have.

---

### Official Review · Reviewer_8sQy · 2023-08-03

**Soundness:** 3

**Excitement:**

3: Ambivalent: It has merits (e.g., it reports state-of-the-art results, the idea is nice), but there are key weaknesses (e.g., it describes incremental work), and it can significantly benefit from another round of revision. However, I won't object to accepting it if my co-reviewers champion it.

**Paper Topic And Main Contributions:**

This paper proposes CONTRASTE, a novel pre-training strategy using contrastive learning to improve the performance of the downstream ASTE task. First, the paper derives aspect-based prompts with corresponding sentiments masked. Then, supervised contrastive learning (SCL) is applied on the decoder-generated aspect-level sentiment embeddings of the masked tokens. The paper demonstrates that the proposed pre-training strategy results in a better downstream performance for ABSA tasks such as ASTE, ACOS, TASD, and AESC, than performing SCL on sentence-level sentiment embeddings. After that, the paper proposes generic placeholder-based templates to fine-tune the encoder-decoder framework. Finally, the paper presents a multi-task model, CONTRASTE-MTL, that achieves SOTA results on the majority of ASTE benchmark datasets. The results demonstrate that the proposed pre-training scheme can improve the performance of other ABSA tasks such as ACOS, TASD, and AESC.

-strengths
1. The problem is cutting edge. This paper studies supervised contrastive pre-training with aspect-based prompts for aspect sentiment triplet extraction, which has been less explored.
2. The supervised contrastive pre-training with aspect-based prompts for aspect sentiment triplet extraction is somewhat novel and the method is well-designed.
3. The narrative and logic of the paper are very clear.
4. Figures are clear, and the design of these figures is nice.
5. The generic placeholder-based templates have greater universality.
6. The proposed pre-training strategy can achieve state-of-the-art results for other ABSA tasks.
7. The experiments are extensive and rigorous, and the explanation of the experiments is detailed.
8. The related works are adequate.
9. The experimental results have high reliability.
10. The paper is well-written and structured.
11. The limitations are frank.

-weaknesses
1. The related works are not well organized.
2. The paper lacks a comparative analysis with ChatGPT.
3. Pre-training a large language model is extremely resource-consuming, and the method proposed in this paper cannot be easily, quickly, and cheaply applied to other situations.
4. Pre-training large language models may not be sufficient, because the amount of training data is not enough.

**Questions For The Authors:**

1. As mentioned in the paper, authors respectively fine-tune the pre-trained model weights for the downstream ASTE task using AdamW optimizer with a learning rate of 1e-4 for 14Res and 16Res, and 3e-4 for 15Res and Lap14. Why the learning rates are different? Do I have to readjust the learning rate if I use a different dataset?
2. As mentioned in the paper, authors start by optimizing α with β set to 0.4, then optimize β given the optimal α values. Why do authors determine the values of α and β in this way? The reason for this is not explained in the paper.

**Reasons To Accept:**

This paper proposes CONTRASTE, a novel pre-training strategy using contrastive learning to improve the performance of the downstream ASTE task. The extensive experiments and detailed analysis also support the proposed method. This work is beneficial to the researchers who study the ASTE task, the ABSA tasks, supervised contrastive learning, etc. Please also refer to the “strengths” for more details.

**Reasons To Reject:**

The paper lacks a comparative analysis with ChatGPT. Additionally, pre-training large language models may not be sufficient, because the amount of training data is not enough. Please also refer to the “weaknesses” for more details.

**Reproducibility:**

4: Could mostly reproduce the results, but there may be some variation because of sample variance or minor variations in their interpretation of the protocol or method.

**Reviewer Confidence:**

4: Quite sure. I tried to check the important points carefully. It's unlikely, though conceivable, that I missed something that should affect my ratings.

---

> ### Author Rebuttal · Authors · 2023-08-29
>
> Thank you for your time and effort in giving us your valuable feedback. We are extremely happy to note that you have generously appreciated the novelty of our proposed pre-training strategy, and its utility in achieving state-of-the-art results for multiple ABSA tasks. Thank you once again for appreciating the detailed narrative of our proposed method, the universality of the generic placeholder-based templates, and the reliability, rigor, and explanation of our experimental results, among other things. Please find below our clarifications/answers to your queries:
>
> 1. "The related works are not well organized."
>
> We note that you have acknowledged the adequacy of the related works as one of the strengths, however we understand your concern. We accept that we could not organize the entire related works section within the main draft owing to space constraints. As already pointed out by you, the novelty of our proposed approach primarily lies in how we apply supervised contrastive pre-training with carefully-curated aspect-based prompts in order to improve multiple ABSA tasks. Accordingly, we organized the prior studies on the application of supervised contrastive learning for ABSA within the main draft. The ASTE baselines are detailed in Section A.3 in the appendix. Similarly, the baselines for the tasks of ACOS, TASD, and AESC are respectively detailed in appendix sections A.4, A.5, and A.6. We pledge to organize the related works section better in the camera-ready draft by utilizing the extra page we will get.
>
> 2. "The paper lacks a comparative analysis with ChatGPT."
>
> Thank you for this suggestion. We carried out exhaustive experiments to understand the capability of ChatGPT in solving multiple ABSA tasks. The ChatGPT version used by us is gpt-3.5-turbo. The “temperature” parameter was set to 0 in order to avoid variations in ChatGPT-generated outputs which is a known fact. This further helps us to reproduce the results in the future with the same settings. For generating the responses, the “max_tokens” parameter was set to 512. We experimented with 4 different prompts to get the response for each test set sentence. We take the example of ASTE to explain our experimental settings:
>
> - Setting 1: Full Zero-shot with no task definition. Here the prompt that we use is:
>
> Given the restaurant/laptop review: {sent}, find all the (aspect, opinion, sentiment) triplets appearing in the review in json format, with review and triplets as the keys. No explanations needed and do not provide any text in the response.
>
> - Setting 2: Zero-shot with task definition. Here, we simply prepend the following task definition for ASTE to the prompt mentioned above:
>
> Aspects are nouns or phrases appearing in the text that indicate specific attributes of the entity being reviewed. Opinion terms are  adjectives or phrases that express specific sentiments towards the aspects. Sentiments could be either positive, negative or neutral.
>
> - Setting 3: Few-shot In-Context-Learning (ICL) with the same training set samples.
>
> To design the prompt in this setting, after the task definition mentioned in Setting 2, and before the final instruction mentioned in Setting 1, we add 100 randomly selected (Sentence, Target Output) samples from the corresponding training set. Please note that the same 100 samples are used as part of the prompt for each test set sentence.
>
> - Setting 4: Few-shot ICL with dynamically selected training set samples:
>
> Here, for each test set sentence, we use a different set of randomly selected 100 samples from the corresponding training set as part of the prompt. While no model parameters are updated in our experiments with ChatGPT, our goal in this setting is to see whether covering a broader range of training set samples helps to improve scores.
>
> Following are the ASTE results with ChatGPT on the benchmark ASTE-Data-V2 datasets:
>
> | Setting       | 14Res |  |  | 15Res |  |  | 16Res |  |  | Lap14 |  |  |
> |---|:---:|:---:|:---:|:---:|:---:|:---:|:---:|:---:|:---:|:---:|:---:|:---:|
> |    | P | R | F1 | P | R | F1 | P | R | F1 | P | R | F1 |
> | 1 | 0.151 | 0.175 | 0.162 | 0.125 | 0.181 | 0.148 | 0.184 | 0.255 | 0.214 | 0.171 | 0.233 | 0.197 |
> | 2 | 0.445 | 0.529 | 0.484 | 0.377 | 0.551 | 0.447 | 0.426 | 0.609 | 0.502 | 0.276 | 0.375 | 0.318 |
> | 3 | 0.513 | 0.629 | 0.565 | 0.419 | 0.606 | 0.495 | 0.449 | 0.617 | 0.520 | 0.351 | 0.489 | 0.409 |
> | 4 | 0.488 | 0.619 | 0.546 | 0.400 | 0.575 | 0.472 | 0.473 | 0.656 | 0.549 | 0.351 | 0.475 | 0.404 |
> | ChatGPT 5-shot ICL results for ASTE reported in https://arxiv.org/pdf/2305.14450.pdf  | - | - | 0.549 | - | - | 0.466 | - | - | 0.518 | - | - | 0.390 |
> | Our best-reported results with CONTRASTE-MTL | 0.736 | 0.744 | 0.740 | 0.653 | 0.667 | 0.661 | 0.722 | 0.763 | 0.742 | 0.642 | 0.617 | 0.629 |
> |
>
>
> We observe that across all 4 datasets, using the task definition helps to improve scores considerably. Prompting with training set examples further improves the scores. However, averaged across all 4 datasets, we do not observe any advantage of Setting 4 over Setting 3. This is expected since the model does not remember the training set examples it was prompted with earlier while generating the response for a new test set sentence. Its memory is limited to the context of the prompt.
>
> Interestingly, an arxiv paper https://arxiv.org/pdf/2305.14450.pdf also reports results for different information extraction tasks using ChatGPT, including ABSA. Please refer to Table 1 of this arxiv paper. For your convenience, we have reported their ASTE results in the table above. We observe that our Setting 3/4 results comfortably outperforms them. Finally, we also compare the ChatGPT results (Setting 3) with our results with CONTRASTE-MTL reported in the paper, and find a huge gap in performance. This confirms the findings of the arxiv paper, and demonstrates that ChatGPT is not equipped enough to give us state-of-the-art results for ASBA tasks.
>
> With necessary task-specific adjustments made to Setting 3, here are the ChatGPT results (F1-scores) for the other ABSA tasks reported in the paper (in comparison to our model scores):
> Please note that the results are obtained on the test sets of respective task-specific datasets.
>
> | Model        | 14Res | 15Res | 16Res | Lap14 | Rest-ACOS | Laptop-ACOS |
> |---|:---:|:---:|:---:|:---:|:---:|:---:|
> | ACOS-ChatGPT | -     |  0.262 | 0.386 | -     |    0.391   | 0.237       |
> | ACOS-Contra  | -     | 0.478 | 0.598 | -     | 0.605     | 0.446       |
> | TASD-ChatGPT | -     | 0.469 | 0.561 | -     | -         | -           |
> | TASD-Contra  | -     | 0.664 | 0.747 | -     | -         | -           |
> | AESC-ChatGPT | 0.634 | 0.556 | 0.623 | 0.541 | -         | -           |
> | AESC-Contra  | 0.826 | 0.741 | 0.810 | 0.731 | -         | -           |
> |
>
> Finally, we pledge to include these results in the camera-ready draft.
>
> 3. "Pre-training a large language model is extremely resource-consuming..."
>
> We understand your concern. However, we would like to clarify a few things. First, please note that we are not pre-training the T5 model from scratch which is an extremely resource and time-consuming task. We start from the pre-trained T5 checkpoint downloaded from the Huggingface library and continue pre-training it using our proposed strategy to improve its aspect-level sentiment understanding before we start fine-tuning the model for respective ABSA tasks. In essence, our proposed method can also be viewed as a two-step fine-tuning process, where the first step is a one-time process where we train the model using contrastive learning, and the second step is to take the updated model weights and further fine-tune them for respective ABSA tasks. Second, please note that all our experiments are performed with T5-base which is a relatively smaller T5 variant with 220M parameters (T5-XXL as 11B parameters). Finally, for the first step, we do not use any external data for performing contrastive pre-training. We limit our experiments to the data obtained from the training sets of our task-specific ABSA datasets, as described in the Section 3.2, thereby avoiding any additional overhead. Hence, we believe our proposed method can be efficiently applied to other situations.
>
> 4. "Pre-training large language models may not be sufficient.."
>
> True. However, we would like to consider this as an advantage of our proposed pre-training technique. As detailed in Section 5 (line 565), prior works on applying supervised contrastive learning for ABSA usually rely on external large-scale sentiment-annotated corpora to learn sentiment knowledge. We, on the other hand, efficiently pre-train our model by only effectively leveraging the limited data from the training sets of benchmark ABSA datasets. We believe that we can further improve our reported results if we make use of additional resources for large-scale pre-training. We shall include these clarifications in the camera-ready draft.
>
> 5. Query regarding different learning rates for different datasets.
>
> Yes. Please note from Table 3 that all 4 ASTE benchmark datasets have different no. of data points, and different distributions of positive, negative, and neutral sentiment triplets. Further, each of them have their own pre-defined train-validation-test splits. While fine-tuning the contrastive pre-trained T5 model weights on the respective datasets, we tuned our hyperparameters on the corresponding validation sets. Please note that hyperparameter tuning on a held-out validation/development set is a standard procedure used across papers. We reported the hyperparameters that helped us to obtain the best triplet-F1 scores on the respective validation sets. In our case, the best learning rates were different for different datasets as reported in Section 3.2. Finally, it may not be always required to readjust the learning rate unless you take a dataset with very different characteristics to work on.
>
> 6. Query regarding setting the α with β parameters.
>
> Please refer to line 338 under Section 3.2 where we have referred to the paper which motivated us to include the OTD module and train our final model, CONTRASTE-MTL, in a multi-task setup. Similar to our case, the referred paper also trains their final model by optimizing a joint loss involving the loss functions for the main task and two auxiliary tasks. We take a strategy similar to the one used in the referred paper to select the auxiliary task weights. We start by optimizing α with β set to 0.4. We then optimize β given the optimal α values. During our experiments, we verified that reversing the order did not change the optimal values. Another strategy we could have used is a grid-search to choose the optimal alpha and beta values simultaneously, however it is a computationally expensive process. We assure you to include a detailed explanation of our parameter-tuning strategies in the camera-ready draft.
>
> Overall, we are happy to address your concerns/suggestions, and would like to answer any follow-up questions that you may have. Also, we would sincerely request you to kindly reconsider your scores, if you are satisfied with our explanations and additional results.

---

### Official Review · Reviewer_ZRNq · 2023-08-10

**Soundness:** 2

**Excitement:**

3: Ambivalent: It has merits (e.g., it reports state-of-the-art results, the idea is nice), but there are key weaknesses (e.g., it describes incremental work), and it can significantly benefit from another round of revision. However, I won't object to accepting it if my co-reviewers champion it.

**Paper Topic And Main Contributions:**

The paper proposes CONTRASTE-MTL, a novel pre-training strategy using contrastive learning to improve the performance on ASTE task.

**Reasons To Accept:**

1. Detailed illustration in both text and picture about the proposed method, which I can follow easily.
2. Comprehensive and promising experiment results.

**Reasons To Reject:**

1. The motivation of the paper is unclear or not strong. 'different from existing work' could not be a good motivation.

2. The mechnism of OTD is unclear. OTD is a token-level task while the generative ASTE is formulated as a sequence-to-sequence generation one, so basically they should not have an mutual influence.
Besides, I also note that in the reference paper as your motivation for add OTD, the target sequence of the two tasks are the same, but it is not the case in this paper's setting, which makes readers more doubtful about the effectiveness of OTD module (although in your ablative results it really works a bit).

3. The paper lacks a deeper analysis into the TCE module, though ablation results show it benefits the model's performance. I am curious about how it really works. My points are as follows:
1) It is a regressor instead of classifier, meaning it could produce float numbers. In this case, how the model could treat a float number output (e.g., 2.4, the decoder intends to stop after producing 2 or 3 aspect term) at inference time?
2) How much chance it can predict the correct aspect term number during inference time?
3) If training with a randomly-generated pesudo labels regardless of its correctness can still enhance the performance.

**Reproducibility:**

3: Could reproduce the results with some difficulty. The settings of parameters are underspecified or subjectively determined; the training/evaluation data are not widely available.

**Reviewer Confidence:**

4: Quite sure. I tried to check the important points carefully. It's unlikely, though conceivable, that I missed something that should affect my ratings.

---

> ### Author Rebuttal · Authors · 2023-08-29
>
> Thank you for your time and effort in giving us your valuable feedback. We are happy to note that you have appreciated our detailed illustration of the proposed method and comprehensive experimental results. Please find below our clarifications/answers to your queries:
>
> 1. "The motivation of the paper is unclear..."
>
> We understand your concern. Please note from line 45 that all existing works on ASTE have predominantly tried to develop newer, and often more complex fine-tuning techniques for the task. Our primary motivation for the work, as mentioned on line 48, is however to design an efficient pre-training strategy to enhance the aspect-aware sentiment understanding of our proposed model. This helps us to improve the downstream performances not only for ASTE but also for other ABSA tasks, such as ACOS, TASD, and AESC, as demonstrated in the paper. As detailed in Section 5, the novelty of our work further lies in the way we apply contrastive learning on aspect-level sentiment embeddings of masked opinion words (obtained using carefully-curated aspect-based prompts to guide the decoder) instead of sentence-level embeddings. Overall, what makes our approach different from existing works is that we propose a generic pre-training technique to improve multiple ABSA tasks simultaneously rather than focusing on more sophisticated task-specific fine-tuning techniques for respective ABSA tasks. We shall clarify these points more clearly in the camera-ready draft (if accepted).
>
> 2. "The mechanism of OTD is unclear..."
>
> Thank you for highlighting your concern, however we would beg to differ on the following two grounds. First, in the reference paper (https://aclanthology.org/2022.findings-acl.156.pdf), which motivated us to include the OTD module, please note from Section 3.1 (in the referred paper) that the main task of “Autoregressive Entity Linking” is formulated as a “language generation task”, whereas the auxiliary subtask of “Entity Mention Detection” (refer Section 3.2) is modeled as a token-wise binary classification task. Similarly we formulate ASTE, our main task, as a sequence-to-sequence generative task, whereas the auxiliary subtask of OTD is formulated as a sequence-tagging task using the BIO scheme, thereby making it a token-level 3-way classification task as described on line 274.
>
> Second, as mentioned on line 266, we hypothesize that OTD will help to better detect the opinion span boundaries which in turn should help to improve ASTE (mainly in opinion term extraction and sentiment prediction). As already acknowledged by you, in Table 6 (ablation results), we report that the inclusion of OTD module helps to improve the Opinion-F1 scores by 2.4% (0.848 vs. 0.828), Sentiment-Accuracy scores by 1.8% (0.784 vs. 0.770), and finally the ASTE/Triplet-F1 scores by 1% (0.692 vs. 0.685). Also, please note from line 301 that this mutual influence results from the fact that we are training our model (CONTRASTE-MTL) in a multi-task setup, jointly optimizing the generative, OTD, and TCE loss functions.
>
> 3. "The paper lacks a deeper analysis into the TCE module...". We sequentially provide answers to the three concerns raised by you:
>
> - 3.1. "It is a regressor instead of classifier..."
>
> First of all, we would like to clarify that the number of triplets varies across sentences. Also, a test set sentence may well contain more triplets than any other sentence it has seen during training. Hence, we modeled the auxiliary subtask of triplet count estimation (TCE) as a regressor instead of a classifier since we do not know the exact no. of triplets apriori. Please note from line 287, that TCE is introduced to implicitly guide the decoder when to stop. As mentioned before, this implicit guidance is expected through the joint optimization of loss functions for the three modules, encoder-decoder, OTD and TCE. Please note that there is no explicit connection between the output of the TCE module and the decoder. As mentioned on line 283, the explicit guidance for the decoder on how many triplets to generate is provided through supervision from ground truth target sequences.
>
> We agree that the TCE module can produce float numbers. During training, we drive the model behavior by minimizing the Mean Squared Error loss as mentioned on line 292. During inference, we round off the output to the nearest integer (and 2.5 to 2). Therefore, for a TCE-generated value of 2.4, we would ideally want the decoder to stop after generating 2 triplets. However, please note again that there is no such direct influence. We also do not post-process the decoder-generated sequence to match the same number of triplets as predicted by the TCE module before calculating the reported results.
>
> - 3.2. "How much chance it can predict the correct aspect term number during inference time?"
>
> Please note that the motivation behind the inclusion of the TCE module was to improve the model’s capability to predict the **correct no. of triplets** for a given sentence, and **not the correct no. of aspect terms**. The same aspect may be repeated across several triplets. For example, the sentence “the food was absolutely fresh and delicious, however overpriced” contains three opinion triplets as follows: (food, absolutely fresh, positive), (food, delicious, positive), and (food, overpriced, negative). Here, all three triplets contain the same aspect term “food”. The inclusion of the TCE module however improves, not only the overall ASTE performance as already acknowledged by you, but also the Aspect-F1 scores by 1% (0.840 vs. 0.832) (referring to the ablation results reported in Table 6). We however attribute this observation to the improved capacity of the model to generate the correct no. of triplets, which in turn directly impacts all the scores.
>
> Following your suggestions, we tried to quantify the performance of the TCE module separately and also to what extent it matches with the actual no. of triplets generated by the decoder. Please note that for calculating the results, we round off the TCE-generated float values to the nearest integers, as mentioned above. The results are reported in the table below:
>
> | Dataset                                                                                          | 14Res | 15Res | 16Res | Lap14 |
> |----------------------------------------------------------------------------------------------------------------------|-------|-------|-------|-------|
> | TCE Accuracy                                                                                                         | 92\%  | 88\%  | 90\%  | 85\%  |
> | Percentage of times the no. of triplets predicted by TCE matches the actual no. of triplets generated by the decoder | 93\%  | 93\%  | 95\%  | 92\%  |
> |
>
> We observe that the TCE module is quite accurate, predicting the correct no. of triplets close to 89% of the times, averaged across all 4 ASTE datasets. Further, in more than 93% of the cases on average, the no. of triplets predicted by the TCE module matches the actual no. of triplets generated by the decoder, thereby demonstrating the synergy between the two components.
>
>
> - 3.3. "If training with a randomly-generated pesudo labels.."
>
> We observe from above that the TCE module is fairly accurate, and that the TCE and decoder predict the same no. of triplets in more than 93% of the cases. If the TCE module is trained with randomly-generated pseudo labels, we expect a negative impact on the overall ASTE scores, despite accepting the fact that the TCE and decoder do not have any explicit influence apart from their losses being optimized together.  In the best scenario, the model can learn to ignore this random signal, but then the overall ASTE performance would be at best as reported without the TCE module (second last row in Table 6).
>
> We are happy to address your concerns/suggestions, and would like to answer any follow-up questions that you may have. However, please note that our main contribution is to propose a novel pre-training strategy that would improve multiple ABSA tasks rather than focusing on a specific ABSA task. From Table 5, we observe that ASTE-Base has a triplet F1 score of 0.671 averaged across all 4 datasets, which improves to 0.680 (CONTRASTE-Base) with our proposed contrastive learning-based pre-training technique. Training the model in a multi-task setup (CONTRASTE-MTL), with OTD and TCE modules included, helps us to outperform Span-BiDir, our strongest baseline, by a margin of 1% in triplet F1 scores (0.692 vs. 0.685). Finally, from Tables 8, 9, 10, and 11, we find that our proposed strategy to pre-train an encoder-decoder network by means of applying contrastive learning on aspect-level sentiment embeddings improves all other ASBA tasks.
>
> We pledge to report the suggested clarifications and results in the camera-ready draft (if accepted). In this regard, we sincerely request you to kindly reconsider your scores, if you find our explanations/clarifications satisfactory.

---

### Meta-Review · Area_Chair_VzPZ · 2023-09-20

**Recommendation:** 4

**Metareview:**

The paper proposes a new pre-training strategy using contrastive learning for Aspect Sentiment Triplet Extraction. The experimental setup is sound and thorough. The proposed method is novel and provides good performance gains on the task. The authors should clarify a bit better the efficacy of individual components such as TCE and improve the related work section as well as the motivation behind their approach.

---

### Decision · Program_Chairs · 2023-10-07

**Decision:**

Accept-Findings

**Comment:**

The paper proposes a new pre-training strategy using contrastive learning for Aspect Sentiment Triplet Extraction. The experimental setup is sound and thorough. The proposed method is novel and provides good performance gains on the task. The authors should clarify a bit better the efficacy of individual components such as TCE and improve the related work section as well as the motivation behind their approach.